# Model-based Reinforcement Learning and the Eluder Dimension

**Ian Osband**
Stanford University
iosband@stanford.edu

**Benjamin Van Roy**
Stanford University
bvr@stanford.edu

## Abstract

We consider the problem of learning to optimize an unknown Markov decision process (MDP). We show that, if the MDP can be parameterized within some known function class, we can obtain regret bounds that scale with the dimensionality, rather than cardinality, of the system. We characterize this dependence explicitly as $\tilde{O}(\sqrt{d_K d_E T})$ where $T$ is time elapsed, $d_K$ is the Kolmogorov dimension and $d_E$ is the *eluder dimension*. These represent the first unified regret bounds for model-based reinforcement learning and provide state of the art guarantees in several important settings. Moreover, we present a simple and computationally efficient algorithm *posterior sampling for reinforcement learning* (PSRL) that satisfies these bounds.

## 1 Introduction

We consider the reinforcement learning (RL) problem of optimizing rewards in an unknown Markov decision process (MDP) [1]. In this setting an agent makes sequential decisions within its enironment to maximize its cumulative rewards through time. We model the environment as an MDP, however, unlike the standard MDP planning problem the agent is unsure of the underlying reward and transition functions. Through exploring poorly-understood policies, an agent may improve its understanding of its environment but it may improve its short term rewards by exploiting its existing knowledge [2, 3].

The focus of the literature in this area has been to develop algorithms whose performance will be close to optimal in some sense. There are numerous criteria for statistical and computational efficiency that might be considered. Some of the most common include PAC (Probably Approximately Correct) [4], MB (Mistake Bound) [5], KWIK (Knows What It Knows) [6] and regret [7]. We will focus our attention upon regret, or the shortfall in the agent's expected rewards compared to that of the optimal policy. We believe this is a natural criteria for performance during learning, although these concepts are closely linked. A good overview of various efficiency guarantees is given in section 3 of Li et al. [6].

Broadly, algorithms for RL can be separated as either model-based, which build a generative model of the environment, or model-free which do not. Algorithms of both type have been developed to provide PAC-MDP bounds polynomial in the number of states $S$ and actions $A$ [8, 9, 10]. However, model-free approaches can struggle to plan efficient exploration. The only near-optimal regret bounds to time $T$ of $\tilde{O}(S\sqrt{AT})$ have only been attained by model-based algorithms [7, 11]. But even these bounds grow with the cardinality of the state and action spaces, which may be extremely large or even infinite. Worse still, there is a lower bound $\Omega(\sqrt{SAT})$ for the expected regret in an arbitrary MDP [7].

In special cases, where the reward or transition function is known to belong to a certain functional family, existing algorithms can exploit the structure to move beyond this "'tabula rasa" (where nothing is assumed beyond $S$ and $A$) lower bound. The most widely-studied

parameterization is the degenerate MDP with no transitions, the mutli-armed bandit [12, 13, 14]. Another common assumption is that the transition function is linear in states and actions. Papers here establigh regret bounds $\tilde{O}(\sqrt{T})$ for linear quadratic control [16], but with constants that grow exponentially with dimension. Later works remove this exponential dependence, but only under significant sparsity assumptions [17]. The most general previous analysis considers rewards and transitions that are $\alpha$-Hölder in a $d$-dimensional space to establish regret bounds $\tilde{O}(T^{(2d+\alpha)/(2d+2\alpha)})$ [18]. However, the proposed algorithm UCCRL is not computationally tractable and the bounds approach linearity in many settings.

In this paper we analyse the simple and intuitive algorithm *posterior sampling for reinforcement learning* (PSRL) [20, 21, 11]. PSRL was initially introduced as a heuristic method [21], but has since been shown to satisfy state of the art regret bounds in finite MDPs [11] and also exploit the structure of factored MDPs [15]. We show that this same algorithm satisfies general regret bounds that depends upon the dimensionality, rather than the cardinality, of the underlying reward and transition function classes. To characterize the complexity of this learning problem we extend the definition of the eluder dimension, previously introduced for bandits [19], to capture the complexity of the reinforcement learning problem. Our results provide a unified analysis of model-based reinforcement learning in general and provide new state of the art bounds in several important problem settings.

## 2 Problem formulation

We consider the problem of learning to optimize a random finite horizon MDP $M = (\mathcal{S}, \mathcal{A}, R^M, P^M, \tau, \rho)$ in repeated finite episodes of interaction. $\mathcal{S}$ is the state space, $\mathcal{A}$ is the action space, $R^M(s,a)$ is the reward distribution over $\mathbb{R}$ and $P^M(\cdot|s,a)$ is the transition distribution over $\mathcal{S}$ when selecting action $a$ in state $s$, $\tau$ is the time horizon, and $\rho$ the initial state distribution. All random variables we will consider are on a probability space $(\Omega, \mathcal{F}, \mathbb{P})$.

A policy $\mu$ is a function mapping each state $s \in \mathcal{S}$ and $i = 1, \ldots, \tau$ to an action $a \in \mathcal{A}$. For each MDP $M$ and policy $\mu$, we define a value function $V$:

$$V_{\mu,i}^M(s) := \mathbb{E}_{M,\mu}\Big[ \sum_{j=i}^{\tau} \bar{r}^M(s_j, a_j) \Big| s_i = s \Big] \tag{1}$$

where $\bar{r}^M(s,a) := \mathbb{E}[r|r \sim R^M(s,a)]$ and the subscripts of the expectation operator indicate that $a_j = \mu(s_j, j)$, and $s_{j+1} \sim P^M(\cdot|s_j, a_j)$ for $j = i, \ldots, \tau$. A policy $\mu$ is said to be optimal for MDP $M$ if $V_{\mu,i}^M(s) = \max_{\mu'} V_{\mu',i}^M(s)$ for all $s \in \mathcal{S}$ and $i = 1, \ldots, \tau$. We will associate with each MDP $M$ a policy $\mu^M$ that is optimal for $M$.

We require that the state space $\mathcal{S}$ is a subset of $\mathbb{R}^d$ for some finite $d$ with a $\|\cdot\|_2$-norm induced by an inner product. These result actually extend to general Hilbert spaces, but we will not deal with that in this paper. This allows us to decompose the transition function as a mean value in $\mathcal{S}$ plus additive noise $s' \sim P^M(\cdot|s,a) \implies s' = \bar{p}^M(s,a) + \epsilon_P$. At first this may seem to exclude discrete MDPs with $S$ states from our analysis. However, we can represent the discrete state as a probability vector $s_t \in \mathcal{S} = [0,1]^S \subset \mathbb{R}^S$ with a single active component equal to 1 and 0 otherwise. In fact, the notational convention that $\mathcal{S} \subseteq \mathbb{R}^d$ should not impose a great restriction for most practical settings.

For any distribution $\Phi$ over $\mathcal{S}$, we define the one step future value function $U$ to be the expected value of the optimal policy with the next state distributed according to $\Phi$.

$$U_i^M(\Phi) := \mathbb{E}_{M,\mu^M}\big[ V_{\mu^M, i+1}^M(s) \big| s \sim \Phi \big]. \tag{2}$$

One natural regularity condition for learning is that the future values of similar distributions should be similar. We examine this idea through the Lipschitz constant on the means of these state distributions. We write $\mathcal{E}(\Phi) := \mathbb{E}[s|s \sim \Phi] \in \mathcal{S}$ for the mean of a distribution $\Phi$ and express the Lipschitz continuity for $U_i^M$ with respect to the $\|\cdot\|_2$-norm of the mean:

$$|U_i^M(\Phi) - U_i^M(\tilde{\Phi})| \leq K_i^M(\mathcal{D})\|\mathcal{E}(\Phi) - \mathcal{E}(\tilde{\Phi})\|_2 \text{ for all } \Phi, \tilde{\Phi} \in \mathcal{D} \tag{3}$$

We define $K^M(\mathcal{D}) := \max_i K_i^M(\mathcal{D})$ to be a global Lipschitz contant for the future value function with state distributions from $\mathcal{D}$. Where appropriate, we will condense our notation

to write $K^M := K^M(\mathcal{D}(M))$ where $\mathcal{D}(M) := \{P^M(\cdot|s,a)|s \in \mathcal{S}, a \in \mathcal{A}\}$ is the set of all possible one-step state distributions under the MDP $M$.

The reinforcement learning agent interacts with the MDP over episodes that begin at times $t_k = (k-1)\tau + 1$, $k = 1, 2, \ldots$. Let $H_t = (s_1, a_1, r_1, \ldots, s_{t-1}, a_{t-1}, r_{t-1})$ denote the history of observations made *prior* to time $t$. A reinforcement learning algorithm is a deterministic sequence $\{\pi_k | k = 1, 2, \ldots\}$ of functions, each mapping $H_{t_k}$ to a probability distribution $\pi_k(H_{t_k})$ over policies which the agent will employ during the $k$th episode. We define the regret incurred by a reinforcement learning algorithm $\pi$ up to time $T$ to be

$$\text{Regret}(T, \pi, M^*) := \sum_{k=1}^{\lceil T/\tau \rceil} \Delta_k,$$

where $\Delta_k$ denotes regret over the $k$th episode, defined with respect to the MDP $M^*$ by

$$\Delta_k := \int_{s \in \mathcal{S}} \rho(s) \left( V_{\mu^*,1}^{M^*} - V_{\mu_k,1}^{M^*} \right)(s)$$

with $\mu^* = \mu^{M^*}$ and $\mu_k \sim \pi_k(H_{t_k})$. Note that regret is not deterministic since it can depend on the random MDP $M^*$, the algorithm's internal random sampling and, through the history $H_{t_k}$, on previous random transitions and random rewards. We will assess and compare algorithm performance in terms of regret and its expectation.

## 3 Main results

We now review the algorithm PSRL, an adaptation of Thompson sampling [20] to reinforcement learning. PSRL was first proposed by Strens [21] and later was shown to satisfy efficient regret bounds in finite MDPs [11]. The algorithm begins with a prior distribution over MDPs. At the start of episode $k$, PSRL samples an MDP $M_k$ from the posterior. PSRL then follows the policy $\mu_k = \mu^{M_k}$ which is optimal for this *sampled* MDP during episode $k$.

---

**Algorithm 1**
Posterior Sampling for Reinforcement Learning (PSRL)

---

1: **Input:** Prior distribution $\phi$ for $M^*$, t=1
2: **for** episodes $k = 1, 2, ..$ **do**
3:     sample $M_k \sim \phi(\cdot|H_t)$
4:     compute $\mu_k = \mu^{M_k}$
5:     **for** timesteps $j = 1, .., \tau$ **do**
6:         apply $a_t \sim \mu_k(s_t, j)$
7:         observe $r_t$ and $s_{t+1}$
8:         advance $t = t + 1$
9:     **end for**
10: **end for**

---

To state our results we first introduce some notation. For any set $\mathcal{X}$ and $\mathcal{Y} \subseteq \mathbb{R}^d$ for $d$ finite let $\mathcal{P}_{\mathcal{X},\mathcal{Y}}^{C,\sigma}$ be the family the distributions from $\mathcal{X}$ to $\mathcal{Y}$ with mean $\|\cdot\|_2$-bounded in $[0, C]$ and additive $\sigma$-sub-Gaussian noise. We let $N(\mathcal{F}, \alpha, \|\cdot\|_2)$ be the $\alpha$-covering number of $\mathcal{F}$ with respect to the $\|\cdot\|_2$-norm and write $n_{\mathcal{F}} = \log(8N(\mathcal{F}, 1/T^2, \|\cdot\|_2)T)$ for brevity. Finally we write $d_E(\mathcal{F}) = \dim_E(\mathcal{F}, T^{-1})$ for the eluder dimension of $\mathcal{F}$ at precision $T^{-1}$, a notion of dimension specialized to sequential measurements described in Section 4.

Our main result, Theorem 1, bounds the expected regret of PSRL at any time $T$.

**Theorem 1** (Expected regret for PSRL in parameterized MDPs)**.**
*Fix a state space $\mathcal{S}$, action space $\mathcal{A}$, function families $\mathcal{R} \subseteq \mathcal{P}_{\mathcal{S} \times \mathcal{A}, \mathbb{R}}^{C_{\mathcal{R}}, \sigma_{\mathcal{R}}}$ and $\mathcal{P} \subseteq \mathcal{P}_{\mathcal{S} \times \mathcal{A}, \mathcal{S}}^{C_{\mathcal{P}}, \sigma_{\mathcal{P}}}$ for any $C_{\mathcal{R}}, C_{\mathcal{P}}, \sigma_{\mathcal{R}}, \sigma_{\mathcal{P}} > 0$. Let $M^*$ be an MDP with state space $\mathcal{S}$, action space $\mathcal{A}$, rewards $R^* \in \mathcal{R}$ and transitions $P^* \in \mathcal{P}$. If $\phi$ is the distribution of $M^*$ and $K^* = K^{M^*}$ is a global Lipschitz constant for the future value function as per (3) then:*

$$\mathbb{E}[\text{Regret}(T, \pi^{PS}, M^*)] \leq \left[ C_{\mathcal{R}} + C_{\mathcal{P}} \right] + \tilde{D}(\mathcal{R}) + + \mathbb{E}[K^*] \left( 1 + \frac{1}{T-1} \right) \tilde{D}(\mathcal{P}) \qquad (4)$$

*Where for $\mathcal{F}$ equal to either $\mathcal{R}$ or $\mathcal{P}$ we will use the shorthand:*

$$\tilde{D}(\mathcal{F}) := 1 + \tau C_{\mathcal{F}} d_E(\mathcal{F}) + 8\sqrt{d_E(\mathcal{F})(4C_{\mathcal{F}} + \sqrt{2\sigma_{\mathcal{F}}^2 \log(32T^3)})} + 8\sqrt{2\sigma_{\mathcal{F}}^2 n_{\mathcal{F}} d_E(\mathcal{F})T}.$$

Theorem 1 is a general result that applies to almost all RL settings of interest. In particular, we note that any bounded function is sub-Gaussian. To clarify the assymptotics if this bound we use another classical measure of dimensionality.

**Definition 1.** *The Kolmogorov dimension of a function class $\mathcal{F}$ is given by:*

$$\dim_K(\mathcal{F}) := \limsup_{\alpha \downarrow 0} \frac{\log(N(\mathcal{F}, \alpha, \|\cdot\|_2))}{\log(1/\alpha)}.$$

Using Definition 1 in Theorem 1 we can obtain our Corollary.

**Corollary 1** (Assymptotic regret bounds for PSRL in parameterized MDPs)**.**
*Under the assumptions of Theorem 1 and writing $d_K(\mathcal{F}) := \dim_K(\mathcal{F})$:*

$$\mathbb{E}[\text{Regret}(T, \pi^{PS}, M^*)] = \tilde{O}\left( \sigma_{\mathcal{R}} \sqrt{d_K(\mathcal{R})d_E(\mathcal{R})T} + \mathbb{E}[K^*]\sigma_{\mathcal{P}} \sqrt{d_K(\mathcal{P})d_E(\mathcal{P})T} \right) \quad (5)$$

*Where $\tilde{O}(\cdot)$ ignores terms logarithmic in $T$.*

In Section 4 we provide bounds on the eluder dimension of several function classes. These lead to explicit regret bounds in a number of important domains such as discrete MDPs, linear-quadratic control and even generalized linear systems. In all of these cases the eluder dimension scales comparably with more traditional notions of dimensionality. For clarity, we present bounds in the case of linear-quadratic control.

**Corollary 2** (Assymptotic regret bounds for PSRL in bounded linear quadratic systems)**.**
*Let $M^*$ be an $n$-dimensional linear-quadratic system with $\sigma$-sub-Gaussian noise. If the state is $\|\cdot\|_2$-bounded by $C$ and $\phi$ is the distribution of $M^*$, then:*

$$\mathbb{E}[\text{Regret}(T, \pi^{PS}, M^*)] = \tilde{O}\left(\sigma C \lambda_1 n^2 \sqrt{T}\right). \quad (6)$$

*Here $\lambda_1$ is the largest eigenvalue of the matrix $Q$ given as the solution of the Ricatti equations for the unconstrained optimal value function $V(s) = -s^T Q s$ [22].*

*Proof.* We simply apply the results of for eluder dimension in Section 4 to Corollary 1 and upper bound the Lipschitz constant of the constrained LQR by $2C\lambda_1$, see Appendix D. $\square$

Algorithms based upon posterior sampling are intimately linked to those based upon optimism [14]. In Appendix E we outline an optimistic variant that would attain similar regret bounds but with high probility in a frequentist sense. Unfortunately this algorithm remains computationally intractable even when presented with an approximate MDP planner. Further, we believe that PSRL will generally be more statistically efficient than an optimistic variant with similar regret bounds since the algorithm is not affected by loose analysis [11].

## 4   Eluder dimension

To quantify the complexity of learning in a potentially infinite MDP, we extend the existing notion of eluder dimension for real-valued functions [19] to vector-valued functions. For any $\mathcal{G} \subseteq \mathcal{P}_{\mathcal{X},\mathcal{Y}}^{C,\sigma}$ we define the set of mean functions $\mathcal{F} = \mathbb{E}[\mathcal{G}] := \{f | f = \mathbb{E}[G] \text{ for } G \in \mathcal{G}\}$. If we consider sequential observations $y_i \sim G^*(x_i)$ we can equivalently write them as $y_i = f^*(x_i) + \epsilon_i$ for some $f^*(x_i) = \mathbb{E}[y | y \sim G^*(x_i)]$ and $\epsilon_i$ zero mean noise. Intuitively, the eluder dimension of $\mathcal{F}$ is the length $d$ of the longest possible sequence $x_1, .., x_d$ such that for all $i$, knowing the function values of $f(x_1), .., f(x_i)$ will not reveal $f(x_{i+1})$.

**Definition 2** $((\mathcal{F}, \epsilon) - dependence)$**.**
*We will say that $x \in \mathcal{X}$ is $(\mathcal{F}, \epsilon)$-dependent on $\{x_1, ..., x_n\} \subseteq \mathcal{X}$*

$$\iff \forall f, \tilde{f} \in \mathcal{F}, \quad \sum_{i=1}^{n} \|f(x_i) - \tilde{f}(x_i)\|_2^2 \le \epsilon^2 \implies \|f(x) - \tilde{f}(x)\|_2 \le \epsilon.$$

*$x \in \mathcal{X}$ is $(\epsilon, \mathcal{F})$-independent of $\{x_1, .., x_n\}$ iff it does not satisfy the definition for dependence.*

**Definition 3** (Eluder Dimension).
*The eluder dimension $\dim_E(\mathcal{F}, \epsilon)$ is the length of the longest possible sequence of elements in $\mathcal{X}$ such that for some $\epsilon' \geq \epsilon$ every element is $(\mathcal{F}, \epsilon')$-independent of its predecessors.*

Traditional notions from supervised learning, such as the VC dimension, are not sufficient to characterize the complexity of reinforcement learning. In fact, a family learnable in constant time for supervised learning may require arbitrarily long to learn to control well [19]. The eluder dimension mirrors the linear dimension for vector spaces, which is the length of the longest sequence such that each element is linearly independent of its predecessors, but allows for nonlinear and approximate dependencies. We overload our notation for $\mathcal{G} \subseteq \mathcal{P}_{\mathcal{X}, \mathcal{Y}}^{C;\sigma}$ and write $\dim_E(\mathcal{G}, \epsilon) := \dim_E(\mathbb{E}[\mathcal{G}], \epsilon)$, which should be clear from the context.

## 4.1 Eluder dimension for specific function classes

Theorem 1 gives regret bounds in terms of the eluder dimension, which is well-defined for any $\mathcal{F}, \epsilon$. However, for any given $\mathcal{F}, \epsilon$ actually calculating the eluder dimension may take some additional analysis. We now provide bounds on the eluder dimension for some common function classes in a similar approach to earlier work for real-valued functions [14]. These proofs are available in Appendix C.

**Proposition 1** (Eluder dimension for finite $\mathcal{X}$).
*A counting argument shows that for $|\mathcal{X}| = X$ finite, any $\epsilon > 0$ and any function class $\mathcal{F}$:*

$$\dim_E(\mathcal{F}, \epsilon) \leq X$$

*This bound is tight in the case of independent measurements.*

**Proposition 2** (Eluder dimension for linear functions).
*Let $\mathcal{F} = \{f \mid f(x) = \theta\phi(x) \text{ for } \theta \in \mathbb{R}^{n \times p}, \phi \in \mathbb{R}^p, \|\theta\|_2 \leq C_\theta, \|\phi\|_2 \leq C_\phi\}$ then $\forall \mathcal{X}$:*

$$\dim_E(\mathcal{F}, \epsilon) \leq p(4n-1)\frac{e}{e-1}\log\left[\left(1 + \left(\frac{2C_\phi C_\theta}{\epsilon}\right)^2\right)(4n-1)\right] + 1 = \tilde{O}(np)$$

**Proposition 3** (Eluder dimension for quadratic functions).
*Let $\mathcal{F} = \{f \mid f(x) = \phi(x)^T \theta \phi(x) \text{ for } \theta \in \mathbb{R}^{p \times p}, \phi \in \mathbb{R}^p, \|\theta\|_2 \leq C_\theta, \|\phi\|_2 \leq C_\phi\}$ then $\forall \mathcal{X}$:*

$$\dim_E(\mathcal{F}, \epsilon) \leq p(4p-1)\frac{e}{e-1}\log\left[\left(1 + \left(\frac{2pC_\phi^2 C_\theta}{\epsilon}\right)^2\right)(4p-1)\right] + 1 = \tilde{O}(p^2).$$

**Proposition 4** (Eluder dimension for generalized linear functions).
*Let $g(\cdot)$ be a component-wise independent function on $\mathbb{R}^n$ with derivative in each component bounded $\in [\underline{h}, \overline{h}]$ with $\underline{h} > 0$. Define $r = \frac{\overline{h}}{\underline{h}} > 1$ to be the condition number. If $\mathcal{F} = \{f \mid f(x) = g(\theta\phi(x)) \text{ for } \theta \in \mathbb{R}^{n \times p}, \phi \in \mathbb{R}^p, \|\theta\|_2 \leq C_\theta, \|\phi\|_2 \leq C_\phi\}$ then for any $\mathcal{X}$:*

$$\dim_E(\mathcal{F}, \epsilon) \leq p\left(r^2(4n-2)+1\right)\frac{e}{e-1}\left(\log\left[\left(r^2(4n-2)+1\right)\left(1 + \left(\frac{2C_\theta C_\phi}{\epsilon}\right)^2\right)\right]\right) + 1 = \tilde{O}(r^2 np)$$

## 5 Confidence sets

We now follow the standard argument that relates the regret of an optimistic or posterior sampling algorithm to the construction of confidence sets [7, 11]. We will use the eluder dimension build confidence sets for the reward and transition which contain the true functions with high probability and then bound the regret of our algorithm by the maximum deviation within the confidence sets. For observations from $f^* \in \mathcal{F}$ we will center the sets around the least squares estimate $\hat{f}_t^{LS} \in \arg\min_{f \in \mathcal{F}} L_{2,t}(f)$ where $L_{2,t}(f) := \sum_{i=1}^{t-1} \|f(x_t) - y_t\|_2^2$ is the cumulative squared prediciton error. The confidence sets are defined $\mathcal{F}_t = \mathcal{F}_t(\beta_t) := \{f \in \mathcal{F} \mid \|f - \hat{f}_t^{LS}\|_{2,E_t} \leq \sqrt{\beta_t}\}$ where $\beta_t$ controls the growth of the confidence set and the empirical 2-norm is defined $\|g\|_{2,E_t}^2 := \sum_{i=1}^{t-1} \|g(x_i)\|_2^2$.

For $\mathcal{F} \subseteq \mathcal{P}_{\mathcal{X},\mathcal{Y}}^{C,\sigma}$, we define the distinguished control parameter:

$$\beta_t^*(\mathcal{F}, \delta, \alpha) := 8\sigma^2 \log(N(\mathcal{F}, \alpha, \|\cdot\|_2)/\delta) + 2\alpha t \left(8C + \sqrt{8\sigma^2 \log(4t^2/\delta)}\right) \quad (7)$$

This leads to confidence sets which contain the true function with high probability.

**Proposition 5** (Confidence sets with high probability).
*For all $\delta > 0$ and $\alpha > 0$ and the confidence sets $\mathcal{F}_t = \mathcal{F}_t(\beta_t^*(\mathcal{F}, \delta, \alpha))$ for all $t \in \mathbb{N}$ then:*

$$\mathbb{P}\left(f^* \in \bigcap_{t=1}^{\infty} \mathcal{F}_t\right) \geq 1 - 2\delta$$

*Proof.* We combine standard martingale concentrations with a discretization scheme. The argument is essentially the same as Proposition 6 in [14], but extends statements about $\mathbb{R}$ to vector-valued functions. A full derivation is available in the Appendix A. $\qquad \square$

## 5.1 Bounding the sum of set widths

We now bound the deviation from $f^*$ by the maximum deviation within the confidence set.
**Definition 4** (Set widths).
*For any set of functions $\mathcal{F}$ we define the width of the set at $x$ to be the maximum L2 deviation between any two members of $\mathcal{F}$ evaluated at $x$.*

$$w_{\mathcal{F}}(x) := \sup_{\overline{f}, \underline{f} \in \mathcal{F}} \|\overline{f}(x) - \underline{f}(x)\|_2$$

We can bound for the number of large widths in terms of the eluder dimension.
**Lemma 1** (Bounding the number of large widths).
*If $\{\beta_t > 0 \big| t \in \mathbb{N}\}$ is a nondecreasing sequence with $\mathcal{F}_t = \mathcal{F}_t(\beta_t)$ then*

$$\sum_{k=1}^{m} \sum_{i=1}^{\tau} \mathbb{1}\{w_{\mathcal{F}_{t_k}}(x_{t_k+i}) > \epsilon\} \leq \left(\frac{4\beta_T}{\epsilon^2} + \tau\right) \dim_E(\mathcal{F}, \epsilon)$$

*Proof.* This result follows from proposition 8 in [14] but with a small adjustment to account for episodes. A full proof is given in Appendix B. $\qquad \square$

We now use Lemma 1 to control the cumulative deviation through time.
**Proposition 6** (Bounding the sum of widths).
*If $\{\beta_t > 0 \big| t \in \mathbb{N}\}$ is nondecreasing with $\mathcal{F}_t = \mathcal{F}_t(\beta_t)$ and $\|f\|_2 \leq C$ for all $f \in \mathcal{F}$ then:*

$$\sum_{k=1}^{m} \sum_{i=1}^{\tau} w_{\mathcal{F}_{t_k}}(x_{t_k+i}) \leq 1 + \tau C \dim_E(\mathcal{F}, T^{-1}) + 4\sqrt{\beta_T \dim_E(\mathcal{F}, T^{-1})T} \quad (8)$$

*Proof.* Once again we follow the analysis of Russo [14] and strealine notation by letting $w_t = w_{\mathcal{F}_{t_k}}(x_{t_k+i})$ abd $d = \dim_E(\mathcal{F}, T^{-1})$. Reordering the sequence $(w_1, .., w_T) \to (w_{i_1}, .., w_{i_T})$ such that $w_{i_1} \geq .. \geq w_{i_T}$ we have that:

$$\sum_{k=1}^{m} \sum_{i=1}^{\tau} w_{\mathcal{F}_{t_k}}(x_{t_k+i}) = \sum_{t=1}^{T} w_{i_t} \leq 1 + \sum_{i=1}^{T} w_{i_t} \mathbb{1}\{w_{i_t} \geq T^{-1}\}$$

.
By the reordering we know that $w_{i_t} > \epsilon$ means that $\sum_{k=1}^{m} \sum_{i=1}^{\tau} \mathbb{1}\{w_{\mathcal{F}_{t_k}}(x_{t_k+i}) > \epsilon\} \geq t$.
From Lemma 1, $\epsilon \leq \sqrt{\frac{4\beta_T d}{t - \tau d}}$. So that if $w_{i_t} > T^{-1}$ then $w_{i_t} \leq \min\{C, \sqrt{\frac{4\beta_T d}{t - \tau d}}\}$. Therefore,

$$\sum_{i=1}^{T} w_{i_t} \mathbb{1}\{w_{i_t} \geq T^{-1}\} \leq \tau C d + \sum_{t=\tau d+1}^{T} \sqrt{\frac{4\beta_T d}{t - \tau d}} \leq \tau C d + 2\sqrt{\beta_T} \int_0^T \sqrt{\frac{d}{t}} \, dt \leq \tau C d + 4\sqrt{\beta_T d T}$$

$$\square$$

# 6 Analysis

We will now show reproduce the decomposition of expected regret in terms of the Bellman error [11]. From here, we will apply the confidence set results from Section 5 to obtain our regret bounds. We streamline our discussion of $P^M, R^M, V^M_{\mu,i}, U^M_i$ and $\mathcal{T}^M_\mu$ by simply writing $*$ in place of $M^*$ or $\mu^*$ and $k$ in place of $M_k$ or $\mu_k$ where appropriate; for example $V^*_{k,i} := V^{M^*}_{\tilde{\mu}_k,i}$.

The first step in our ananlysis breaks down the regret by adding and subtracting the *imagined* optimal reward of $\mu_k$ under the MDP $M_k$.

$$\Delta_k = \left(V^*_{*,1} - V^*_{k,1}\right)(s_0) = \left(V^*_{*,1} - V^k_{k,1}\right)(s_0) + \left(V^k_{k,1} - V^*_{k,1}\right)(s_0) \tag{9}$$

Here $s_0$ is a distinguished initial state, but moving to general $\rho(s)$ poses no real challenge. Algorithms based upon optimism bound $(V^*_{*,1} - V^k_{k,1}) \leq 0$ with high probability. For PSRL we use Lemma 2 and the tower property to see that this is zero in expectation.

**Lemma 2** (Posterior sampling).
*If $\phi$ is the distribution of $M^*$ then, for any $\sigma(H_{t_k})$-measurable function $g$,*
$$\mathbb{E}[g(M^*)|H_{t_k}] = \mathbb{E}[g(M_k)|H_{t_k}] \tag{10}$$

We introduce the Bellman operator $\mathcal{T}^M_\mu$, which for any MDP $M = (\mathcal{S}, \mathcal{A}, R^M, P^M, \tau, \rho)$, stationary policy $\mu : \mathcal{S} \to \mathcal{A}$ and value function $V : \mathcal{S} \to \mathbb{R}$, is defined by

$$\mathcal{T}^M_\mu V(s) := \bar{r}^M(s, \mu(s)) + \int_{s' \in \mathcal{S}} P^M(s'|s, \mu(s))V(s').$$

This returns the expected value of state $s$ where we follow the policy $\mu$ under the laws of $M$, for one time step. The following lemma gives a concise form for the dynamic programming paradigm in terms of the Bellman operator.

**Lemma 3** (Dynamic programming equation).
*For any MDP $M = (\mathcal{S}, \mathcal{A}, R^M, P^M, \tau, \rho)$ and policy $\mu : \mathcal{S} \times \{1, \ldots, \tau\} \to \mathcal{A}$, the value functions $V^M_\mu$ satisfy*
$$V^M_{\mu,i} = \mathcal{T}^M_{\mu(\cdot,i)} V^M_{\mu,i+1} \tag{11}$$
*for $i = 1 \ldots \tau$, with $V^M_{\mu,\tau+1} := 0$.*

Through repeated application of the dynamic programming operator and taking expectation of martingale differences we can mirror earlier analysis [11] to equate expected regret with the cumulative Bellman error:

$$\mathbb{E}[\Delta_k] = \sum_{i=1}^{\tau} (\mathcal{T}^k_{k,i} - \mathcal{T}^*_{k,i}) V^k_{k,i+1}(s_{t_k+i}) \tag{12}$$

## 6.1 Lipschitz continuity

Efficient regret bounds for MDPs with an infinite number of states and actions require some regularity assumption. One natural notion is that nearby states might have similar optimal values, or that the optimal value function function might be Lipschitz. Unfortunately, any discontinuous reward function will usually lead to discontious values functions so that this assumption is violated in many settings of interest. However, we only require that the *future* value is Lipschitz in the sense of equation (3). This will will be satisfied whenever the underlying value function is Lipschitz, but is a strictly weaker requirement since the system noise helps to smooth future values.

Since $\mathcal{P}$ has $\sigma_P$-sub-Gaussian noise we write $s_{t+1} = \bar{p}^M(s_t, a_t) + \epsilon^P_t$ in the natural way. We now use equation (12) to reduce regret to a sum of set widths. To reduce clutter and more closely follow the notation of Section 4 we will write $x_{k,i} = (s_{t_k+i}, a_{t_k+i})$.

$$\begin{aligned}
\mathbb{E}[\Delta_k] &\leq \mathbb{E}\left[\sum_{i=1}^{\tau} \left\{\bar{r}^k(x_{k,i}) - \bar{r}^*(x_{k,i}) + U^k_i(P^k(x_{k,i})) - U^k_i(P^*(x_{k,i}))\right\}\right] \\
&\leq \mathbb{E}\left[\sum_{i=1}^{\tau} \left\{|\bar{r}^k(x_{k,i}) - \bar{r}^*(x_{k,i})| + K^k\|\bar{p}^k(x_{k,i}) - \bar{p}^*(x_{k,i})\|_2\right\}\right] \tag{13}
\end{aligned}$$

Where $K^k$ is a global Lipschitz constant for the future value function of $M_k$ as per (3).

We now use the results from Sections 4 and 5 to form the corresponding confidence sets $\mathcal{R}_k := \mathcal{R}_{t_k}(\beta^*(\mathcal{R}, \delta, \alpha))$ and $\mathcal{P}_k := \mathcal{P}_{t_k}(\beta^*(\mathcal{P}, \delta, \alpha))$ for the reward and transition functions respectively. Let $A = \{R^*, R_k \in \mathcal{R}_k \ \forall k\}$ and $B = \{P^*, P_k \in \mathcal{P}_k \ \forall k\}$ and condition upon these events to give:

$$
\begin{aligned}
\mathbb{E}[\text{Regret}(T, \pi^{PS}, M^*)] &\leq \mathbb{E}\left[\sum_{k=1}^{m}\sum_{i=1}^{\tau}\left\{|\bar{r}^k(x_{k,i}) - \bar{r}^*(x_{k,i})| + K^k\|\bar{p}^k(x_{k,i}) - \bar{p}^*(x_{k,i})\|_2\right\}\right] \\
&\leq \sum_{k=1}^{m}\sum_{i=1}^{\tau}\left\{w_{\mathcal{R}_k}(x_{k,i}) + \mathbb{E}[K^k|A,B]w_{\mathcal{P}_k}(x_{k,i}) + 8\delta(C_{\mathcal{R}} + C_{\mathcal{P}})\right\} \quad (14)
\end{aligned}
$$

The posterior sampling lemma ensures that $\mathbb{E}[K^k] = \mathbb{E}[K^*]$ so that $\mathbb{E}[K^k|A,B] \leq \frac{\mathbb{E}[K^*]}{\mathbb{P}(A,B)} \leq \frac{\mathbb{E}[K^*]}{1-8\delta}$ by a union bound on $\{A^c \cup B^c\}$. We fix $\delta = 1/8T$ to see that:

$$
\mathbb{E}[\text{Regret}(T, \pi^{PS}, M^*)] \leq (C_{\mathcal{R}} + C_{\mathcal{P}}) + \sum_{k=1}^{m}\sum_{i=1}^{\tau}w_{\mathcal{R}_k}(x_{k,i}) + \mathbb{E}[K^*]\left(1 + \frac{1}{T-1}\right)\sum_{k=1}^{m}\sum_{i=1}^{\tau}w_{\mathcal{P}_t}(x_{k,i})
$$

We now use equation (7) together with Proposition 6 to obtain our regret bounds. For ease of notation we will write $d_E(\mathcal{R}) = \dim_E(\mathcal{R}, T^{-1})$ and $d_E(\mathcal{P}) = \dim_E(\mathcal{P}, T^{-1})$.

$$
\begin{aligned}
\mathbb{E}[\text{Regret}(T, \pi^{PS}, M^*)] &\leq 2 + (C_{\mathcal{R}} + C_{\mathcal{P}}) + \tau(C_{\mathcal{R}}d_E(\mathcal{R}) + C_{\mathcal{P}}d_E(\mathcal{P})) + \\
&\quad 4\sqrt{\beta_T^*(\mathcal{R}, 1/8T, \alpha)d_E(\mathcal{R})T} + 4\sqrt{\beta_T^*(\mathcal{P}, 1/8T, \alpha)d_E(\mathcal{P})T} \quad (15)
\end{aligned}
$$

We let $\alpha = 1/T^2$ and write $n_{\mathcal{F}} = \log(8N(\mathcal{F}, 1/T^2, \|\cdot\|_2)T)$ for $\mathcal{R}$ and $\mathcal{P}$ to complete our proof of Theorem 1:

$$
\mathbb{E}[\text{Regret}(T, \pi^{PS}, M^*)] \leq [C_{\mathcal{R}} + C_{\mathcal{P}}] + \tilde{D}(\mathcal{R}) + \mathbb{E}[K^*]\left(1 + \frac{1}{T-1}\right)\tilde{D}(\mathcal{P}) \quad (16)
$$

Where $\tilde{D}(\mathcal{F})$ is shorthand for $1 + \tau C_{\mathcal{F}}d_E(\mathcal{F}) + 8\sqrt{d_E(\mathcal{F})(4C_{\mathcal{F}} + \sqrt{2\sigma_{\mathcal{F}}^2\log(32T^3)})} + 8\sqrt{2\sigma_{\mathcal{F}}^2 n_{\mathcal{F}} d_E(\mathcal{F})T}$. The first term $[C_{\mathcal{R}} + C_{\mathcal{P}}]$ bounds the contribution from missed confidence sets. The cost of learning the reward function $R^*$ is bounded by $\tilde{D}(\mathcal{R})$. In most problems the remaining contribution bounding transitions and lost future value will be dominant. Corollary 1 follows from the Definition 1 together with $n_{\mathcal{R}}$ and $n_{\mathcal{P}}$.

## 7 Conclusion

We present a new analysis of *posterior sampling for reinforcement learning* that leads to a general regret bound in terms of the dimensionality, rather than the cardinality, of the underlying MDP. These are the first regret bounds for reinforcement learning in such a general setting and provide new state of the art guarantees when specialized to several important problem settings. That said, there are a few clear shortcomings which we do not address in the paper. First, we assume that it is possible to draw samples from the posterior distribution exactly and in some cases this may require extensive computational effort. Second, we wonder whether it is possible to extend our analysis to learning in MDPs without episodic resets. Finally, there is a fundamental hurdle to model-based reinforcement learning that planning for the optimal policy even in a *known* MDP may be intractable. We assume access to an approximate MDP planner, but this will generally require lengthy computations. We would like to examine whether similar bounds are attainable in model-free learning [23], which may obviate complicated MDP planning, and examine the computational and statistical efficiency tradeoffs between these methods.

### Acknowledgments

Osband is supported by Stanford Graduate Fellowships courtesy of PACCAR inc. This work was supported in part by Award CMMI-0968707 from the National Science Foundation.

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
