[Supplementary Material]

# A Confidence sets with high probability

In this appendix we will build up to a proof of Proposition 5, that the confidence sets defined by $\beta^*$ in equation 7 hold with high probability. We begin with some elementary results from martingale theory.

**Lemma 4** (Exponential Martingale).
*Let $Z_i \in L^1$ be real-calued random variables adapted to $\mathcal{H}_i$. We define the conditional mean $\mu_i = \mathbb{E}[Z_i | \mathcal{H}_{i-1}]$ and conditional cumulant generating function $\psi_i(\lambda) = \log \mathbb{E}[\exp(\lambda(Z_i - \mu_i)) | \mathcal{H}_{i-1}]$, then*

$$M_n(\lambda) = \exp\left(\sum_1^n \lambda(Z_i - \mu_i) - \psi_i(\lambda)\right)$$

*is a martingale with $\mathbb{E}[M_n(\lambda)] = 1$.*

**Lemma 5** (Concentration Guarantee).
*For $Z_i$ adapted real $L^1$ random variables adapted to $\mathcal{H}_i$. We define the conditional mean $\mu_i = \mathbb{E}[Z_i | \mathcal{H}_{i-1}]$ and conditional cumulant generating function $\psi_i(\lambda) = \log \mathbb{E}[\exp(\lambda(Z_i - \mu_i)) | \mathcal{H}_{i-1}]$.*

$$\mathbb{P}\left(\bigcup_{n=1}^{\infty}\{\sum_1^n \lambda(Z_i - \mu_i) - \psi_i(\lambda) \geq x\}\right) \leq e^{-x}$$

Both of these lemmas are available in earlier discussion for real-valued variables [14]. We now specialize our discussion to the vector space $\mathcal{Y} \subseteq \mathbb{R}^d$ where the inner product $<y, y> = \|y\|_2^2$. To simplify notation we will write $f_t^* := f^*(x_t)$ and $f_t = f(x_t)$ for arbitrary $f \in \mathcal{F}$. We now define

$$
\begin{aligned}
Z_t &= \|f_t^* - y_t\|_2 - \|f_t - y_t\|_2 \\
&= <f_t^* - y_t, f_t^* - y_t> - <f_t - y_t, f_t - y_t, f_t - y_t> \\
&= -<f_t - f_t^*, f_t - f_t^*> + 2<f_t - f_t^*, y_t - f_t^*> \\
&= -\|f_t - f_t^*\|_2 + 2<f_t - f_t^*, \epsilon_t>
\end{aligned}
$$

so that clearly $\mu_t = -\|f_t - f_t^*\|_2$. Now since we have said that the noise is $\sigma$-sub-Gaussian, $\mathbb{E}[\exp(<\phi, \epsilon>)] \leq \exp\left(\frac{\|\phi\|_2^2 \sigma^2}{2}\right) \forall \phi \in \mathcal{Y}$. From here we can deduce that:

$$
\begin{aligned}
\psi_t(\lambda) &= \log \mathbb{E}[\exp(\lambda(Z_t - \mu_t)) | \mathcal{H}_{t-1}] \\
&= \log \mathbb{E}[\exp(2\lambda <f_t - f_t^*, \epsilon_t>)] \\
&\leq \frac{\|2\lambda(f_t - f_t^*)\|_2^2 \sigma^2}{2}.
\end{aligned}
$$

We now write $\sum_{i=1}^{t-1} Z_i = L_{2,t}(f^*) - L_{2,t}(f)$ according to our earlier definition of $L_{2,t}$. We can apply Lemma 5 with $\lambda = 1/4\sigma^2$, $x = log(1/\delta)$ to obtain:

$$\mathbb{P}\{\left(L_{2,t}(f) \geq L_{2,t}(f^*) + \frac{1}{2}\|f - f^*\|_{2,E_t} - 4\sigma^2 \log(1/\delta)\right) \forall t\} \geq 1 - \delta$$

substituting $f = \hat{f}$ to be the least squares solution which minimizes $L_{2,t}(f)$ we can remove $L_{2,t}(\hat{f}) - L_{2,t}(f^*) \leq 0$. From here we use an $\alpha$-cover discretization argument to complete the proof of Proposition 5.

Let $\mathcal{F}^\alpha \subset \mathcal{F}$ be an $\alpha$-2 cover of $\mathcal{F}$ such that $\forall f \in \mathcal{F}$ there is some $\|f^\alpha - f\|_2 \leq \alpha$. We can use a union bound on $\mathcal{F}^\alpha$ so that $\forall f \in \mathcal{F}$:

$$L_{2,t}(f) - L_{2,t}(f^*) \geq \frac{1}{2}\|f - f^*\|_{2,E_t} - 4\sigma^2 \log(N(\mathcal{F}, \alpha, \|\cdot\|_2)/\delta) + DE(\alpha) \tag{17}$$

$$\text{For } DE(\alpha) = \min_{f^\alpha \in \mathcal{F}^\alpha}\left\{\frac{1}{2}\|f^\alpha - f^*\|_{2,E_t}^2 - \frac{1}{2}\|f - f^*\|_{2,E_t}^2 + L_{2,t}(f) - L_{2,t}(f^\alpha)\right\}$$

We will now seek to bound this discretization error with high probability.

**Lemma 6** (Bounding discretization error).
*If $\|f^\alpha(x) - f(x)\|_2 \leq \alpha$ for all $x \in \mathcal{X}$ then with probability at least $1 - \delta$:*

$$DE(\alpha) \leq \alpha t \left[8C + \sqrt{8\sigma^2 \log(4t^2/\delta)}\right]$$

*Proof.* For non-trivial bounds we will consider the case of $\alpha \leq C$ and note that via Cauchy-Schwarz:

$$\|f^\alpha(x)\|_2^2 - \|f(x)\|_2^2 \leq \max_{\|y\|_2 \leq \alpha} \|f(x) + y\|_2^2 - \|f\|_2^2 \leq 2C\alpha + \alpha^2.$$

From here we can say that

$$\|f^\alpha(x) - f^*(x)\|_2^2 - \|f(x) - f^*(x)\|_2^2 = \|f^\alpha(x)\|_2^2 - \|f(x)\|_2^2 + 2 < f^*(x), f(x) - f^\alpha(x) > \leq 4C\alpha$$

$$\|y - f(x)\|_2^2 - \|y - f^\alpha(x)\|_2^2 = 2 < y, f^\alpha(x) - f(x) > + \|f(x)\|_2^2 - \|f^\alpha(x)\|_2^2 \leq 2\alpha|y| + 2C\alpha + \alpha^2$$

Summing these expressions over time $i = 1, .., t-1$ and using sub-gaussian high probability bounds on $|y|$ gives our desired result. □

Finally we apply Lemma 6 to equation 17 and use the fact that $\hat{f}_t^{LS}$ is the $L_{2,t}$ minimizer to obtain the result that with probability at least $1 - 2\delta$:

$$\|\hat{f}_t^{LS} - f^*\|_{2,E_t} \leq \sqrt{\beta_t^*(\mathcal{F}, \alpha, \delta)}$$

Which is our desired result.

# B   Bounding the number of large widths

**Lemma 1** (Bounding the number of large widths).
*If $\{\beta_t > 0 \,|\, t \in \mathbb{N}\}$ is a nondecreasing sequence with $\mathcal{F}_t = \mathcal{F}_t(\beta_t)$ then*

$$\sum_{k=1}^m \sum_{i=1}^\tau \mathbb{1}\{w_{\mathcal{F}_{t_k}}(x_{t_k+i}) > \epsilon\} \leq \left(\frac{4\beta_T}{\epsilon^2} + \tau\right) \dim_E(\mathcal{F}, \epsilon)$$

*Proof.* We first imagine that $w_{\mathcal{F}_t}(x_t) > \epsilon$ and is $\epsilon$-dependent on $K$ disjoint subsequences of $x_1, .., x_{t-1}$. If $x_t$ is $\epsilon$-dependent on $K$ disjoint subsequences then there exist $\|\overline{f} - \underline{f}\|_{2,E_t} > K\epsilon^2$. By the triangle inequality $\|\overline{f} - \underline{f}\|_{2,E_t} \leq 2\sqrt{\beta_t} \leq 2\sqrt{\beta_T}$ so that $K < 4\beta_T/\epsilon^2$.

In the case without episodic delay, Russo went on to show that in any sequence of length $l$ there is some element which is $\epsilon$-dependent on at least $\frac{l}{\dim_E(\mathcal{F},\epsilon)} - 1$ disjoint subsequences [14]. Our analysis follows similarly, but we may lose up to $\tau - 1$ proper subsequences due to the delay in updating the episode. This means that we can only say that $K \geq \frac{l}{\dim_E(\mathcal{F},\epsilon)} - \tau$. Considering the subsequence $w_{\mathcal{F}_{t_k}}(x_{t_k+i}) > \epsilon$ we see that $l \leq \left(\frac{4\beta_T}{\epsilon^2} + \tau\right) \dim_E(\mathcal{F}, \epsilon)$ as required. □

# C   Eluder dimension for specific function classes

In this section of the appendix we will provide bounds upon the eluder dimension for some canonical function classes. Recalling Definition 3, $\dim_E(\mathcal{F}, \epsilon)$ is the length $d$ of the longest sequence $x_1, .., x_d$ such that for some $\epsilon' \geq \epsilon$:

$$w_k = \sup\left\{ \|(\overline{f} - \underline{f})(x_k)\|_2 \;\middle|\; \|\overline{f} - \underline{f}\|_{2,E_t} \leq \epsilon' \right\} > \epsilon' \tag{18}$$

for each $k \leq d$.

## C.1   Finite domain $\mathcal{X}$

Any $x \in \mathcal{X}$ is $\epsilon$-dependent upon itself for all $\epsilon > 0$. Therefore for all $\epsilon > 0$ the eluder dimension of $\mathcal{F}$ is bounded by $|\mathcal{X}|$.

## C.2   Linear functions $f(x) = \theta\phi(x)$

Let $\mathcal{F} = \{f \,|\, f(x) = \theta\phi(x)$ for $\theta \in \mathbb{R}^{n \times p}, \phi \in \mathbb{R}^p, \|\theta\|_2 \leq C_\theta, \|\phi\|_2 \leq C_\phi\}$. To simplify our notation we will write $\phi_k = \phi(x_k)$ and $\theta = \theta_1 - \theta_2$. From here, we may manipulate the expression

$$\|\theta\phi\|_2^2 = \phi_k^T \theta^T \theta \phi_k = Tr(\phi_k^T \theta^T \theta \phi_k) = Tr(\theta\phi_k\phi_k^T\theta)$$

$$\implies w_k = \sup_\theta \{\|\theta\phi_k\|_2 \;\big|\; Tr(\theta\Phi_k\theta^T) \leq \epsilon^2\} \text{ where } \Phi_k := \sum_{i=1}^{k-1} \phi_i\phi_i^T$$

We next require a lemma which gives an upper bound for trace constrained optimizations.

**Lemma 7** (Bounding norms under trace constraints).
Let $\theta \in \mathbb{R}^{n \times p}, \phi \in \mathbb{R}^p$ and $V \in \mathbb{R}_{++}^{p \times p}$, the set of positive definite $p \times p$ matrices, then:

$$W^2 = \max_\theta \|\theta\phi\|_2^2 \text{ subject to } Tr(\theta V \theta^T) \leq \epsilon^2$$

is bounded above by $W^2 \leq (2n-1)\epsilon^2 \|\phi\|_{V^{-1}}^2$ where $\|\phi\|_A^2 := \phi^T A \phi$.

*Proof.* We first note that $\|\theta\phi\|_2^2 = Tr(\theta\phi\phi^T\theta^T) = \sum_1^n (\theta\phi)_i^2 \leq \left(\sum_1^n (\theta\phi)_i\right)^2$ by Jensen's inequality. We define $\tilde\Phi \in \mathbb{R}^{n \times p}$ such that each row of $\tilde\Phi = \phi^T$. Then this inequality can be expressed as:

$$W^2 = Tr(\theta\phi\phi^T\theta^T) \leq Sum(\theta \otimes \tilde\Phi)^2$$

Where $A \otimes B = C$ for $C_{ij} = A_{ij}B_{ij}$ and $Sum(C) := \sum_{i,j} C_{ij}$ We can now obtain an upper bound for our original problem through this convex relaxation:

$$\max_\theta Sum(\theta \otimes \tilde\Phi) \text{ subject to } Tr(\theta V \theta^T) \leq \epsilon^2$$

We can now form the lagrangian $\mathcal{L}(\theta, \lambda) = -Sum(\theta \otimes \tilde\Phi) + \lambda(Tr(\theta V \theta^T) - \epsilon^2)$. Solving for first order optimality $\nabla_\theta \mathcal{L} = 0 \implies \theta^* = \frac{1}{2\lambda}\tilde\Phi V^{-1}$. From here we form the dual objective

$$g(\lambda) = -Sum(\frac{1}{2\lambda}\tilde\Phi V^{-1} \otimes \tilde\Phi) + Tr(\frac{1}{4\lambda}\tilde\Phi V^{-1}\tilde\Phi^T) - \lambda\epsilon^2$$

Here we solve for the dual-optimal $\lambda^*$ $\nabla_\lambda g = 0 \implies \frac{1}{2\lambda^*}^2 Sum(\frac{1}{2\lambda}\tilde\Phi V^{-1} \otimes \tilde\Phi) - \frac{1}{4\lambda^*}^2 Tr(\frac{1}{4\lambda}\tilde\Phi V^{-1}\tilde\Phi^T) = \epsilon^2$. From the definition of $\tilde\Phi$, $Sum(\tilde\Phi V^{-1} \otimes \tilde\Phi) = n\phi^T V^{-1}\phi$ and $Tr(\tilde\Phi V^{-1}\tilde\Phi^T) = \phi^T V^{-1}\phi$. From this we can simplify our expression to conclude:

$$\frac{n}{2\lambda^{*2}}\phi^T V^{-1}\phi - \frac{1}{4\lambda^{*2}}\phi^T V^{-1}\phi = \epsilon^2 \implies \lambda^* = \sqrt{\frac{(n-1/2)}{2\epsilon^2}}\|\phi\|_{V^{-1}}$$

$$\implies g(\lambda^*) = -\frac{n}{2\lambda^*}\|\phi\|_{V^{-1}}^2 + \frac{1}{4\lambda^*}\|\phi\|_{V^{-1}}^2 - \lambda^*\epsilon$$

$$\text{strong duality} \implies f(\theta^*) = g(\lambda^*) = \sqrt{2n-1}\epsilon\|\phi\|_{V^{-1}}$$

From here we conclude that the optimal value of $W^2 \leq f(\theta^*)^2 \leq (2n-1)\epsilon^2\|\phi\|_{V^{-1}}^2$. $\qquad\square$

Using this lemma, we will be able to address the eluder dimension for linear functions. Using the definition of $w_k$ from equation 18 together with $\Phi_k$ we may rewrite:

$$w_k = \max_\theta \{\sqrt{Tr(\theta\phi_k\phi_k^T\theta)} \mid Tr(\theta\Phi_k\theta^T) \leq \epsilon^2\}.$$

Let $V_k := \Phi_k + \left(\frac{\epsilon}{2C_\theta}\right)^2 I$ so that $Tr(\theta\Phi_k\theta^T) \leq \epsilon^2 \implies Tr(\theta V_k\theta^T) \leq 2\epsilon^2$ through a triangle inequality. Now applying Lemma 7 we can say that $w_k \leq \epsilon\sqrt{4n-2}\|\phi_k\|_{V_k^{-1}}$. This means that if $w_k \geq \epsilon$ then $\|\phi_k\|_{V_k^{-1}}^2 > \frac{1}{4n-2} > 0$.

We now imagine that $w_i \geq \epsilon$ for each $i < k$. Then since $V_k = V_{k-1} + \phi_k\phi_k^T$ we can use the Matrix Determinant together with the above observation to say that:

$$det(V_k) = det(V_{k-1})(1 + \phi_k^T V_K^{-1}\phi_k) \geq det(V_{k-1})\left(1 + \frac{1}{4n-2}\right) \geq .. \geq \lambda^p\left(1 + \frac{1}{4n-2}\right)^{k-1} \quad (19)$$

for $\lambda := \left(\frac{\epsilon}{2C_\theta}\right)^2$. To get an upper bound on the determinant we note that $det(V_k)$ is maximized when all eigenvalues are equal or equivalently:

$$det(V_k) \leq \left(\frac{Tr(V_k)}{p}\right)^p \leq \left(\frac{C_\phi^2(k-1)}{p} + \lambda\right)^p \quad (20)$$

Now using equations 19 and 20 together we see that $k$ must satisftfy the inequality $\left(1 + \frac{1}{4n-2}\right)^{(k-1)/p} \leq \frac{C_\phi^2(k-1)}{\lambda p} + 1$. We now write $\zeta_0 = \frac{1}{4n-2}$ and $\alpha_0 = \frac{C_\phi^2}{\lambda} = \left(\frac{2C_\phi C_\theta}{\epsilon}\right)^2$ so that we can epress this as:

$$(1 + \zeta_0)^{\frac{k-1}{p}} \leq \alpha_0\frac{k-1}{p} + 1$$

We now use the result that $B(x, \alpha) = \max\{B \mid (1+x)^B \leq \alpha B+1\} \leq \frac{1+x}{x} \frac{e}{e-1} \{\log(1+\alpha) + \log(\frac{1+x}{x})\}$.
We complete our proof of Proposition 2 through computing this upper bound at $(\zeta_0, \alpha_0)$,

$$\dim_E(\mathcal{F}, \epsilon) \leq p(4n-1) \frac{e}{e-1} \log\left[\left(1 + \left(\frac{2C_\phi C_\theta}{\epsilon}\right)^2\right)(4n-1)\right] + 1 = \tilde{O}(np).$$

## C.3  Quadratic functions $f(x) = \phi^T(x)\theta\phi(x)$

Let $\mathcal{F} = \{f \mid f(x) = \phi(x)^T \theta \phi(x)$ for $\theta \in \mathbb{R}^{p \times p}, \phi \in \mathbb{R}^p, \|\theta\|_2 \leq C_\theta, \|\phi\|_2 \leq C_\phi\}$ then for any $\mathcal{X}$ we can say that:

$$\dim_E(\mathcal{F}, \epsilon) \leq p(4p-1) \frac{e}{e-1} \log\left[\left(1 + \left(\frac{2pC_\phi^2 C_\theta}{\epsilon}\right)^2\right)(4p-1)\right] + 1 = \tilde{O}(p^2).$$

Where we have simply applied the linear result with $\tilde{\epsilon} = \frac{\epsilon}{pC_\mathcal{P}}$. This is valid since if we can identify the linear function $g(x) = \theta\phi(x)$ to within this tolerance then we will certainly know $f(x)$ as well.

## C.4  Generalized linear models

Let $g(\cdot)$ be a component-wise independent function on $\mathbb{R}^n$ with derivative in each component bounded $\in [\underline{h}, \overline{h}]$ with $\underline{h} > 0$. Define $r = \frac{\overline{h}}{\underline{h}} > 1$ to be the condition number. If $\mathcal{F} = \{f \mid f(x) = g(\theta\phi(x))$ for $\theta \in \mathbb{R}^{n \times p}, \phi \in \mathbb{R}^p, \|\theta\|_2 \leq C_\theta, \|\phi\|_2 \leq C_\phi\}$ then for any $\mathcal{X}$:

$$\dim_E(\mathcal{F}, \epsilon) \leq p\left(r^2(4n-2)+1\right) \frac{e}{e-1} \left(\log\left[\left(r^2(4n-2)+1\right)\left(1 + \left(\frac{2C_\theta C_\phi}{\epsilon}\right)^2\right)\right]\right) + 1 = \tilde{O}(r^2 np)$$

This proof follows exactly as per the linear case, but first using a simple reduction on the form of equation (18).

$$
\begin{aligned}
w_k &= \sup\left\{ \|(\overline{f} - \underline{f})(x_k)\|_2 \;\Big|\; \|\overline{f} - \underline{f}\|_{2, E_t} \leq \epsilon' \right\} \\
&\leq \max_{\theta_1, \theta_2}\left\{ \|g(\theta_1 \phi_k) - g(\theta_2 \phi_k)\|_2 \;\Big|\; \sum_{i=1}^{k-1} \|g(\theta_1 \phi_i) - g(\theta_2 \phi_i)\|_2^2 \leq \epsilon^2 \right\} \\
&\leq \max_{\theta}\left\{ \overline{h}\|\theta\phi_k\|_2 \;\Big|\; \sum_{i=1}^{k-1} \underline{h}^2 \|\theta\phi_i\|_2^2 \leq \epsilon^2 \right\}
\end{aligned}
$$

To which we can now apply Lemma 7 with the $\epsilon$ rescaled by $r$. Following the same arguments as for linear functions now completes our proof.

# D  Bounded LQR control

We imagine a standard linear quadratic controller with rewards with $x = (s, a)$ the state-action vector. The rewards and transitions are given by:

$$R(x) = -x^T A x + \epsilon_R \;, \; P(x) = \Pi_C(Bx + \epsilon_P),$$

where $A \succeq 0$ is positive semi-definite and $\Pi_C$ projects x onto the $\|\cdot\|_2$-ball at radius $C$.

In the case of unbounded states and actions the Ricatti equations give the form of the optimal value function $V(s) = -s^T Q s$ for $Q \succeq 0$. In this case we can see that the difference in values of two states:

$$|V(s) - V(s')| = |-s^T Q s + s'^T Q s'| = |-(s + s')^T Q(s - s')| \leq 2C\lambda_1 \|s - s'\|_2$$

where $\lambda_1$ is the largest eigenvalue of $Q$ and $C$ is an upper bound on the $\|\cdot\|_2$-norm of both $s$ and $s'$. We note that $2C\lambda_1$ works as an effective Lipshcitz constant when we know what $C$ can bound $s, s'$.

We observe that for any projection $\Pi_C(x) = \alpha x$ for $\alpha \in (0, 1]$ and that for all positive semi-definite matrices $A \succeq 0$, $(\alpha x)^T A(\alpha x) = \alpha^2 x^T A x \leq x^T A x$. Using this observation together with reward and transition functions we can see that the value function of the bounded LQR system is always greater than or equal to that of the unconstrained value function. The effect of excluding the low-reward outer region, but maintaining the higher-reward inner region means that the value function becomes more flat in the bounded case, and so $2C\lambda_1$ works as an effective Lipschitz constant for this problem too.

# E  UCRL-Eluder

For completeness, we explicitly outline an optimistic algorithm which uses the confidence sets in our analysis of PSRL to guarantee similar regret bounds with high probability over all MDP $M^*$. The algorithm follows the style of UCRL2 [7] so that at the start of the $k$th episode the algorithm form $\mathcal{M}_k = \{M | P^M \in \mathcal{P}_k, R^M \in \mathcal{R}_k\}$ and then solves for the optimistic policy that attains the highest reward over any $M$ in $\mathcal{M}_k$.

---

**Algorithm 2**
UCRL-Eluder

---
1: **Input:**  Confidence parameter $\delta > 0$, t=1
2: **for** episodes $k = 1, 2, ..$ **do**
3:     form confidence sets $\mathcal{R}_k(\beta^*(\mathcal{R}, \delta, 1/k^2))$ and $\mathcal{P}_k(\beta^*(\mathcal{P}, \delta, 1/k^2))$
4:     compute $\mu_k$ optimistic policy over $\mathcal{M}_k = \{M | P^M \in \mathcal{P}_k, R^M \in \mathcal{R}_k\}$
5:     **for** timesteps $j = 1, .., \tau$ **do**
6:         apply $a_t \sim \mu_k(s_t, j)$
7:         observe $r_t$ and $s_{t+1}$
8:         advance $t = t + 1$
9:     **end for**
10: **end for**

---

Generally, step 4 of this algorithm with not be computationally tractable even when solving for $\mu^M$ is possible for a given $M$.