[Reviews · NeurIPS 2014]

Submitted by Assigned_Reviewer_1

Summary: This paper provides a Bayesian expected regret bound for the Posterior Sampling for the Reinforcement Learning (PSRL) algorithm. PSRL has been introduced by [Strens2000], and can be seen as the application of Thompson sampling for RL problems: a model is sampled from the (posterior) distribution over models, the optimal policy for the sampled model is calculated, the policy is followed until the end of the horizon, and the distribution over models is updated.
PSRL for finite MDPs has been analyzed by [OVRR2013], but the main contribution of this paper is to analyze PSRL for MDPs with general state and action space. In the analysis, the authors use the concept of eluder dimension introduced by [RVR2013]. Eluder dimension was previously used in the analysis of bandit problems (for both Thompson Sampling and the Optimism in Face of Uncertainty (OFU) approaches). Its application in the analysis of RL problems is novel.

This is a good paper. It addresses the important problem of exploration-exploitation tradeoff in RL and provides a theoretical guarantee for a type of algorithm that is less-studied in the RL/bandits literature (posterior sampling as opposed to the more common OFU type of algorithms).

Quality: The theory seems fine, but there is one issue that should be clarified about the type of provided guarantee. The expected regret guarantee is w.r.t. the prior distribution, i.e., this is a Bayesian guarantee. I think the authors should be more clear and upfront about this.

Clarity: The narrative of the paper, especially the analysis part, can be improved. Section 5 analyzes the set widths without giving much reason why it should be done. This will be cleared a few pages later in Section 6.

Originality: The result is new. The proof techniques are based on [RVR2013] and [OVRR2013], so in some sense these results are not totally "unexpected".

Significance: The problem of exploration-exploitation is important in RL. This paper ensures that one class of approaches to address this problem is theoretically justified.

Other comments:

- L053: I believe the authors meant to refer [AS2011] instead of [13].

- L072: The term "random" in "random finite horizon MDP" should be clarified.

- L094: Using Q as a distribution is confusing for many readers as it is often used to refer to an action-value function.

- L101: What does it mean to take expectation of the difference of two distributions? (|| E[Q - \tilde{Q} || ). In general, I believe the authors should be more clear about their use of expectation symbol.

- L131: Typo: "samples and MDP" ---> "samples an MDP"

- L151: It is not clear what "the family of distributions from X to Y with mean norm bounded …" means. I believe you are referring to a probability kernel from X to Y. I do not understand the mean norm boundedness. Please clarify.

- L152: The error scale used for the covering number is 1/T^2 while the scale used for the eluder dimension is 1/T. I am curious to know why the scale is different, especially since in [RVR2013], both scales are 1/T^2.

- L172, 185, …: Typo: Assymptotic ---> asymptotic

- L210: Typo: extra f (after f(x_i) ).

- L239: What does \forall \mathcal{X} refer to? Isn't the space \mathcal{X} fixed?
- L309: Typo: stealine ---> streamline.

- L310: Typo: abd ---> and

[AS2011] Abbasi-Yadkori and Szepesvari, "Regret Bounds for the Adaptive Control of Linear Quadratic Systems," COLT, 2011.
[Strens2000] Strens, "A Bayesian Framework for Reinforcement Learning," ICML 2000.
[RVR2013] Russo and Van Roy, "Eluder Dimension and the Sample Complexity of Optimistic Exploration," NIPS 2013.
[OVRR2013] Osband, Van Roy, and Russo, "(More) Efficient Reinforcement Learning via Posterior Sampling," NIPS 2013.
Summary: This paper addresses the important problem of exploration-exploitation tradeoff in RL and provides a theoretical guarantee for a type of algorithm that is less-studied in the RL/bandits literature (posterior sampling as opposed to the more common OFU type of algorithms).

Submitted by Assigned_Reviewer_27

The paper presents a general theoretical analysis of model-based reinforcement learning algorithms. The presented regret bounds depend on the Eluder dimensions of the function class that we want to learn for the transition function and the reward function. The eluder dimension captures the complexity of the model by, said intuitively, the number of samples that are needed to learn the real function well. This theoretical analysis is applicable for many model-based RL algorithm, including the commonly used posterior sampling for reinforcement learning (PSRL) algorithm.

Quality:

The theoretical analysis seems to be well done in the paper, although the proofs are very hard to follow or check. While the theoretical contribution seems to be promising, the presentation of the paper is rather poor and I think it is almost impossible to understand the paper in a reasonable amount of time. For more comments on the clarity of the paper see below.

Another maybe not so crucial point is the missing evaluations. I know it is not common for theoretical bounds, but I am always a bit sceptical about these papers where only bounds are presented without any evaluation. You never know how useful the bounds are. I think adding an evaluation of the presented bounds, for example, on a linear quadratic Gaussian (LQG) system would highly upgrade the paper. For example, the authors could evaluate the regret of the PSRL algorithm for randomly sampled LQG problems with different dimensionalities and show the tightness of the bound. I think such an experiment would also help considerably to understand the contribution and it should be easy enough to do.

Significance:

Theoretical bounds for a very general set of model-based RL algorithms can have a high impact in the field. However, it is always the question how tight the bounds are in comparison to the results achieved with the real algorithms. Hence, even just a small experimental evaluation of the algorithm on a toy task would considerably increase the significance of the paper.

Clarity:

-The clarity is the main weakness of the paper. The authors assume too much knowledge of the reader about complex mathematical concepts that are not really common. I think the authors need to concentrate more on transmitting the message of the main result and put more of the very long derivations in the appendix. In the paper it would be more important to explain the intuition and motivation of each step that is important to obtain the bounds.
- I think it is necessary to reduce Equation 4 to its most important parts. I do not think that not all the terms are relevant to understand the main message of the paper. The presented bounds go over two full lines. The authors should spend more time on explaining the more relevant parts of the equation (but put the full equation in the appendix)
- Explaining the Eluder dimension and the Kolmogorov complexity right in the beginning would greatly simplify the understanding of the paper
- define alpha covering and sub-Gaussian noise
- define eluder dimension (section 4) before presenting the main results (section 3)
- For each of the sections, a small description why the following steps are needed would be really helpful
- The paper would also benefit a lot from proofreading. Many sentences are hard to understand due to grammatical mistakes and some of the used words are quite strange (see minor)

Originality:

The presented analysis using the Eluder dimension is a new contribution.

Minor Points:
- line 47: falls victim -> reformulate
- line 48: exploit MDP structure -> the MDP structure
- interedependence -> typo
- "what is more, ..." -> reformulate (additionally, moreover, furthermore...)
- Define distribution Q for Eq 2 in more detail (distribution over what, is it a conditional?)
- 103: "contant"
- Corollary 2: Typically in a continuous linear MDP, the action dimension is smaller than the state dimension (half for dynamical systems). How would that change the result?
- Conclusion: reformulate "what is more". Quagmire is a quite uncommon word
Summary: The paper presents a potentially nice theoretical analysis. However, the paper is lacking clarity such that it is almost impossible to understand the derivations of the bounds. The paper would also benefit from an empirical analysis of the bounds on a small toy problem.

Submitted by Assigned_Reviewer_36

SUMMARY:

The submitted paper extends previous theoretical results in the multi-armed bandit problem to general episodic model-based reinforcement learning in state spaces with an L2-norm. It presents a novel class of regret bounds for an existing exploration algorithm based on an adaptation of Thompson sampling to RL. These regret bounds depend on the complexity of the transition and reward functions rather than the number of states and actions. Complexity is characterized by a metric called eluder dimension, measuring the degree of dependence among actions and rewards. The regret bounds are instantiated for some standard RL settings. Empirical results are not provided.

COMMENTS:

Overall, this is a good paper.

Efficient exploration guarantees which depend on the complexity of the model and not on the size of state and action spaces are one of the most important topics in RL. The presented results provide clear advances into this direction.

The paper employs a lot of math which is hard to grasp at first reading. More explanation and motivation for introduced terms and the individual steps would be helpful.

The paper makes strong use of the theoretical results published in a previously paper, but presents novel and important generalizations to model-based RL.

The limitations are clearly characterized (episodic resets, problem of efficient planning in complex domains, bounds for model-based RL only).

The discussion of related work, however, is not complete. The previous work on KWIK-R-max (e.g., see Li, 2009) is very similar in spirit, though using PAC-MDP bounds. Like the submitted paper, this work establishes results beyond the "tabula rasa setting" for general structured state and action spaces. While the submitted paper makes use of a metric called "eluder dimension" to measure model complexity, KWIK-R-max uses the concept of efficient (KWIK) model learners and is thereby not restricted to state spaces with L2-norm.
Summary: The submitted paper advances model-based reinforcement learning research by presenting novel and interesting regret bounds for an important class of RL settings. It is hard to follow the mathematical details, but besides it is well written and could be accepted.
Author Feedback
Author rebuttal: Thank you very much for your feedback, we will try to deal with the main points in review.

1. Paper clarity
**************************************
The main feedback seems to be that the paper is not easy to follow. After some time away from writing the paper we agree with the reviewers that we did not succeed in making the paper as clear or accessible as we would have liked. We intend to seriously refactor the paper to make the message and intuition more clear and move some of the more technical or tedious analysis to the appendix.

2. Simulation
**************************************
We considered adding simulation results for our algorithms, but did not include them for two reasons. First, we found that in classic linear quadratic control it was very hard to get any learning algorithm that would outperform a simple certain equaivalent policy (for which there are no guarantees). Second, in this theoretical paper with tight page constraints we thought that it might further clutter the message. However, we do agree that some simulation results might be a nice addition to the paper which we should revisit and potentially include as part of our final revision.

3. Discussion of related work
**************************************
We should have included a discussion of KWIK-Rmax in our paper, which deals with a similar problem setting and this was an oversight on our part. However, we believe that our analysis is more general than KWIK-Rmax rather than less. It is our understanding that KWIK-Rmax requires that the state transitions be linear in some kernel representation, which we believe our analysis would cover given knowledge of that kernel. Although we must represent the state in an L2-space, this should not really restrict our analysis. For example, as we show in the paper, we can deal with discrete spaces {1,..,n} (which are not L2) via an embedding in in the L2 space [0,1]^n.